# High-spatial resolution functional chemistry of nitrogen compounds in the observed UK meteorite fall Winchcombe

Christian Vollmer [1] ✉, Demie Kepaptsoglou [2,3], Jan Leitner [4,5], Aleksander B. Mosberg [2], Khalil El Hajraoui [2,3], Ashley J. King [6], Charlotte L. Bays [6,7], Paul F. Schofield [6], Tohru Araki [8,9] & Quentin M. Ramasse [2,10]

Organic matter in extraterrestrial samples is a complex material that might have played an important role in the delivery of prebiotic molecules to the early Earth. We report here on the identification of nitrogen-containing compounds such as amino acids and N-heterocycles within the recent observed meteorite fall Winchcombe by high-spatial resolution spectroscopy techniques. Although nitrogen contents of Winchcombe organic matter are low (N/C ~ 1–3%), we were able to detect the presence of these compounds using a low-noise direct electron detector. These biologically relevant molecules have therefore been tentatively found within a fresh, minimally processed meteorite sample by high spatial resolution techniques conserving the overall petrographic context. Carbon functional chemistry investigations show that sizes of aromatic domains are small and that abundances of carboxylic functional groups are low. Our observations demonstrate that Winchcombe represents an important addition to the collection of carbonaceous chondrites and still preserves pristine extraterrestrial organic matter.

Extraterrestrial samples such as primitive carbonaceous chondrites, chondritic-porous interplanetary dust particles (IDPs), Antarctic micrometeorites (AMMs), or samples returned from asteroids contain up to a few wt% of carbon, of which a significant fraction is organic matter (OM)[1–5]. This OM consists of small (soluble) and large, kerogen-like (insoluble) molecules. Soluble organic matter (SOM) (for example, amino acids or simple alcohols) can be extracted with solvents and analyzed in large quantities by high-resolution mass spectrometers[6–9]. Insoluble organic matter (IOM) can be extracted from the bulk sample by acid dissolution, which results in a demineralized, concentrated organic residue for further studies[10,11]. A third approach that permits

local measurements of soluble and insoluble OM within the original petrographic context is by high-spatial resolution electron microscopy and X-ray spectroscopy studies on samples processed with techniques that are far less invasive than these chemical extraction methods, e.g., raw chips or polished sections, or thin lamellae fabricated by focused ion beam (FIB) tools[12–17]. With this approach, it is possible to investigate the petrographic context of the OM as unaltered as possible and potentially detect both soluble and insoluble components. Hereafter, we will denote these as 'minimally processed' samples.

In addition to sample preparation methods, terrestrial alteration can also modify the functional chemistry and elemental composition

[1]Institut für Mineralogie, Universität Münster, Münster, Germany. [2]SuperSTEM Laboratory, Keckwick Lane, Daresbury, UK. [3]School of Physics, Engineering and Technology, University of York, Heslington, UK. [4]Institut für Geowissenschaften, Ruprecht-Karls-Universität Heidelberg, Heidelberg, Germany. [5]Max Planck Institute for Chemistry, Particle Chemistry Department, Mainz, Germany. [6]Planetary Materials Group, Natural History Museum, London, UK. [7]Department of Earth Sciences, Royal Holloway, University of London, Egham, UK. [8]Diamond Light Source, Harwell Science and Innovation Campus, Didcot, UK. [9]National Institutes of Natural Sciences, Institute for Molecular Science, UVSOR Synchrotron Facility, Okazaki, Japan. [10]School of Chemical and Process Engineering and School of Physics and Astronomy, University of Leeds, Leeds, UK. ✉e-mail: christian.vollmer@uni-muenster.de

of OM in extraterrestrial samples. For example, loss of hydrogen atoms is a well-known secondary process that leads to decreasing H/C ratios of OM, as has been documented in Tagish Lake (C2$_{ung}$) fragments with variable alteration stages[18]. Furthermore, an increasing abundance of nitro ($NO_2$-) or nitrate ($NO_3$-) groups within extraterrestrial OM points to oxidation of nitrogen-containing functional groups[19]. It is therefore of great importance to analyze extraterrestrial OM showing minimal terrestrial overprint, for example within meteorites that were recovered shortly after their observed fall.

The Winchcombe meteorite, classified as a petrologic type 2 Mighei-type (CM2) carbonaceous chondrite, represents such a pristine sample. Its fall over the UK on February 28, 2021 was observed by over a thousand eyewitnesses and is thus one of the most widely recorded carbonaceous chondrite falls to date[20–24]. The main mass of this meteorite (~320 g) was collected~12 h after the fireball observation, with a further 200 g recovered within one week and having experienced no rainfall during this time. Therefore, terrestrial alteration effects in Winchcombe are minimal, although limited modification has been observed on its outer portions[25]. Nevertheless, the Winchcombe meteorite offers a unique opportunity to investigate OM within a CM chondrite from a known location in the solar system and without severe terrestrial alteration effects. Initial studies of OM in bulk Winchcombe samples have confirmed the presence of IOM and SOM, including prebiotic molecules such as amino acids and polycyclic aromatic hydrocarbons[26,27]. However, functional chemistry investigations of OM within minimally processed meteorites such as Winchcombe that have suffered minimal terrestrial alteration are, in general, very sparse.

Typically, high-spatial resolution investigations focus on the carbon functional chemistry of extraterrestrial OM, because it is by far the most abundant element in this material (carbon contents > 90 at. %). It is usually not viable to acquire high signal-to-noise spectra of minor element bonding environments such as nitrogen. However, nitrogen-containing compounds such as N-heterocycles or amino acids are crucial prebiotic molecules and of high interest to understand the potential extraterrestrial origins of life on the early Earth[28–32].

Here we report an investigation of OM grains within the Winchcombe meteorite by high-spatial and -energy resolution electron microscopy and synchrotron spectroscopy techniques, with a focus on nitrogen-bearing compounds. We analyzed the OM within a minimally processed, polished section (Fig. S1) and FIB-prepared lamellae by X-ray Absorption Near-Edge Structure (XANES) in the Scanning Transmission X-ray Microscope (STXM), as well as by using a low-noise direct electron detector attached to an Electron Energy Loss Spectrometer (EELS) in a dedicated, monochromated Scanning Transmission Electron Microscope (STEM). This low-noise detector offers the opportunity to investigate specifically the nitrogen functional chemistry of OM, despite the nitrogen abundances being quite low (N/C ratios in Winchcombe OM are only 1-3%, as confirmed by our EELS analyses). Through our complementary approaches, it was possible to deduce H-C-N-O bonding environments of organic grains with very high spatial resolution (nominally nm to sub-nm for STEM-EELS) as well as ultra-high energy resolution ($\Delta E < 0.1$ eV for both STXM and EELS) and document different types of biologically relevant nitrogen-bearing compounds such as amino acids and nucleobases (*Methods and Supplementary Information*).

## Results and discussion
### Organic matter textures and C K-edge functional chemistry
Almost all organic grains display highly irregular boundaries or diffuse textures with sizes from a few 10 s of nm to several μm in length (Figs. 1 and 2). In general, we observe two types of OM. The larger (>200 nm) grains are more compact with irregular grain boundaries, but clearly distinct from the surrounding phyllosilicate-rich fine-grained matrix. The second fraction occurs as highly diffuse and dispersed material (<200 nm) intermingled with phyllosilicates on a nanometer scale (Figs. 1b and 3). Globular or multi-globular roundish OM, a certain type of OM that is frequently observed in carbonaceous chondrites[2,14,15,17,33–35], is generally rare among the investigated lithologies in this work. We observed a single ~500 nm "nanoglobule" in one lamella, FIBW-05 (Fig. 2c), and a multiglobular organic assemblage in another, FIBW-04.

Synchrotron X-ray absorption spectroscopy analyses of six of the extracted lamellae (FIBW-01 – FIBW-06) shows the OM to have a rather homogeneous functional chemistry at the carbon ionization K-edge (Fig. 4a). Two bands at ~285 eV and at 286.5 eV dominate the spectra of all organic grains. These bands relate to aromatic/olefinic carbon and aromatic ketone/aldehyde (C=O bonding or "carbonyl" in bridging aromatic units) and/or nitrile bonding. Aliphatic (~287.5 eV) and carboxylic COOH- (288.5 eV) bonding that is commonly observed in other meteoritic OM[1,2,14,15] is relatively reduced in these Winchcombe samples. Relative quantification of the "aromaticity" content[17] (Fig. 4b) shows values between $0.69 \pm 0.02$ and $1.22 \pm 0.06$, with an average of 0.94. FIBW-04 has the highest aromaticity value of all the Winchcombe OM grains, while the two FIBW-03 grains and FIBW-06 are almost half that of FIBW-04. This supports the more pristine character of the OM in FIBW-04 and suggests that the irregular grains in FIBW-03 may be more altered by aqueous activity. Quantification of the aromatics / aliphatics and CO + COOH/aliphatics ratios as in Vinogradoff et al.

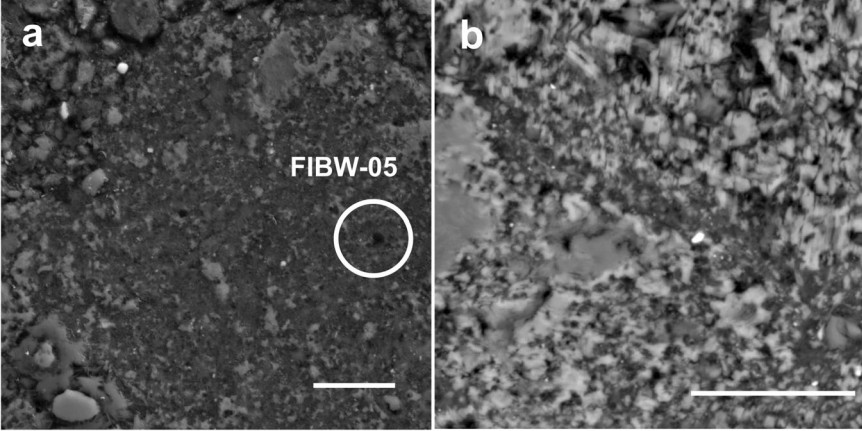

**Fig. 1 | SEM-BSE (back-scattered electron) micrographs at low magnification of typical organic particle morphologies in Winchcombe. a** Irregular to globular organic particle FIBW-05 within the surrounding phyllosilicate-rich matrix. Scale bar is 8 μm. **b** Diffuse organic matter dispersed within the matrix of the more altered lithology B[22] (see Methods and Fig. S1). Scale bar is 10 μm.

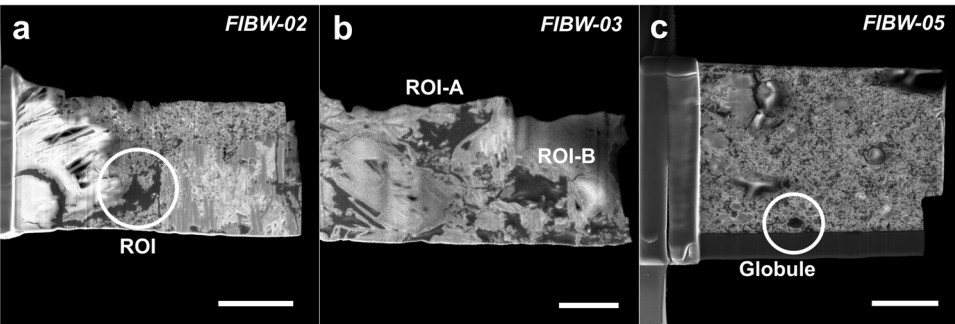

**Fig. 2 | FIB-SEM micrographs at high magnification of extracted lamellae.**
**a** Lamella FIBW-02 with a large, irregular organic grain. The "ROI" defines the area used for STXM analyses. Scale bar is 3 µm. **b** Lamella FIBW-03 containing two large, irregular organic grains. The "ROI-A" and "ROI-B" define the areas used for STXM

analyses. Scale bar is 2 µm. **c** Lamella FIBW-05 showing the extremely fine-grained phyllosilicate-rich texture and the targeted globular organic particle (encircled). Scale bar is 3 µm.

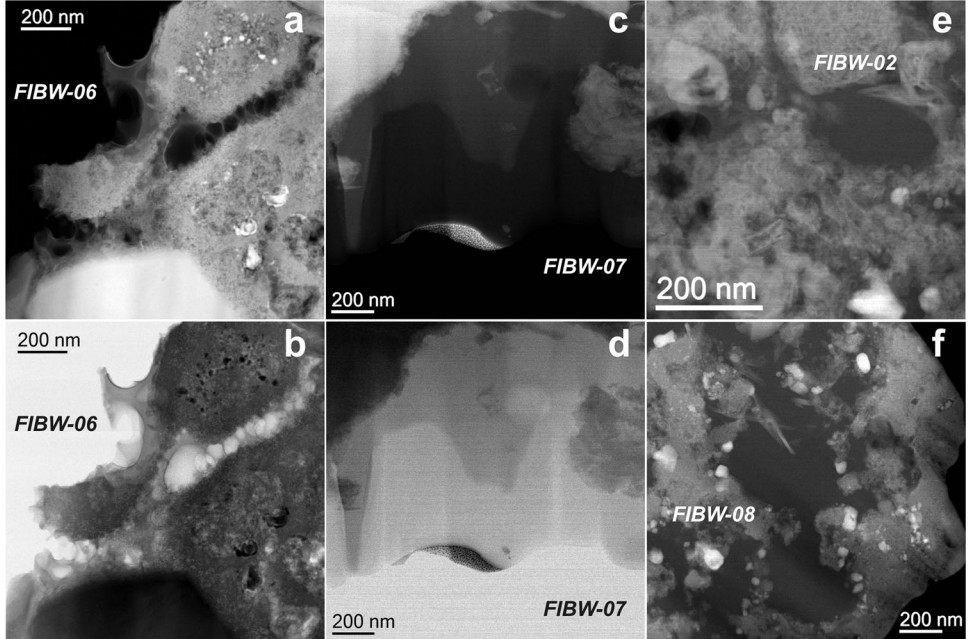

**Fig. 3 | STEM-BF and −HAADF micrographs of typical organic grain morphologies at the higher resolution afforded by the STEM. a, b** Organic matter within FIBW-06 (**a** HAADF, **b** BF) showing irregular and diffuse morphologies. **c, d** Organic matter within FIBW-07 (**c**: HAADF, **d**: BF) showing irregular, but more compact

morphologies. **e** HAADF image of lamella FIBW-02 with a small, globular organic grain. **f** HAADF image of lamella FIBW-08 showing a large and irregular organic grain intermingled with phyllosilicates and other minerals.

(2017)[13] gives higher values than Paris (CM) IOM and Murchison (CM) IOM for our Winchcombe samples (Fig. 5). This most likely implies that abundances of aliphatic branches within the Winchcombe OM is lower than in those two meteorites compared to the other dominating functional groups. It also implies a higher grade of general alteration in comparison to Paris and Murchison, because aliphatic functional groups of OM are easily disturbed by alteration reactions[13,36]. Those two meteorites are usually regarded as less-altered members of the CM chondrite group, with Paris at around petrologic type 2.6–2.9 and Murchison at around 2.5.

The carbon functional chemistry analyzed only by EELS (FIBW07 – FIBW-09) is in general very similar to the STXM data of those lamellae where both techniques have been applied in succession, which provides good confidence that the different approaches are complementary and valid (Figs. 6 and S2-S6). Interestingly, C K-edge spectra obtained by the higher-spatial resolution electron beam revealed localized variability, e.g., with several regions yielding a less pronounced 285 eV "aromaticity" peak than in typical STXM spectra.

The STXM analyses aimed at larger regions of interest (ROIs) that could be easily identified by the broader synchrotron beam, i.e., the more compact type of OM in Winchcombe. In contrast, the much finer electron beam allowed an analysis of smaller more diffuse OM areas within the same lamellae showing less intense absorption in the 285 eV region (Figs. S2–6). Furthermore, in one sample, FIBW-02, we observe a weak but distinct feature at around 287 eV in the EEL spectrum associated with aliphatic bonding (arrow in Fig. 6). In the associated STXM spectrum, this peak is not present, which demonstrates the small-scale variability of this feature in different regions of the lamella, as we might not have aimed at the exact same location in this case. In contrast, the relatively lower intensity of the carboxylic COOH- peak persisted at similar levels in both higher spatial resolution of the STEM compared to the broader synchrotron beam analyses.

## Fingerprints of organic carbon bonding
Organic matter in extraterrestrial samples can be classified based on C K-edge functional chemistry fingerprints[1,2]. These bonding

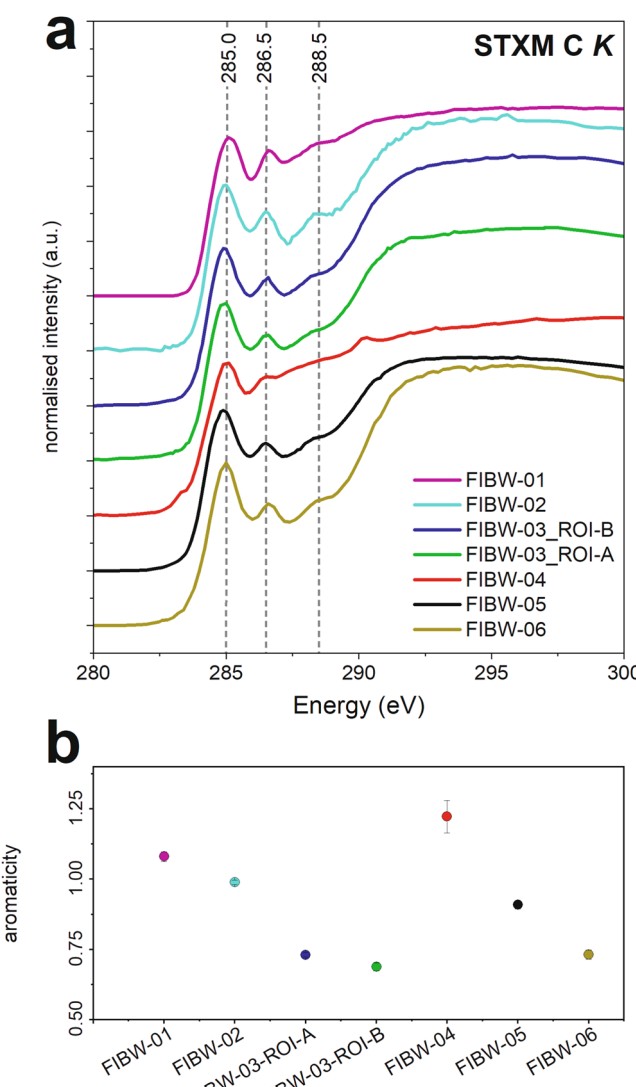

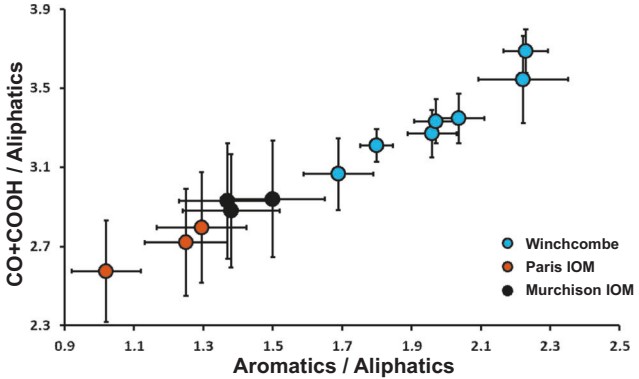

**Fig. 4 | STXM data at the carbon K-edge.** The three main absorption bands of "insoluble organic matter" (IOM) are marked **a**, and the corresponding "aromaticity" contents calculated using the method of Le Guillou et al. (2014)[17] **b**. FIBW-04 contains multiglobular organic matter and FIBW-05 a nanoglobule. Error bars are 1σ.

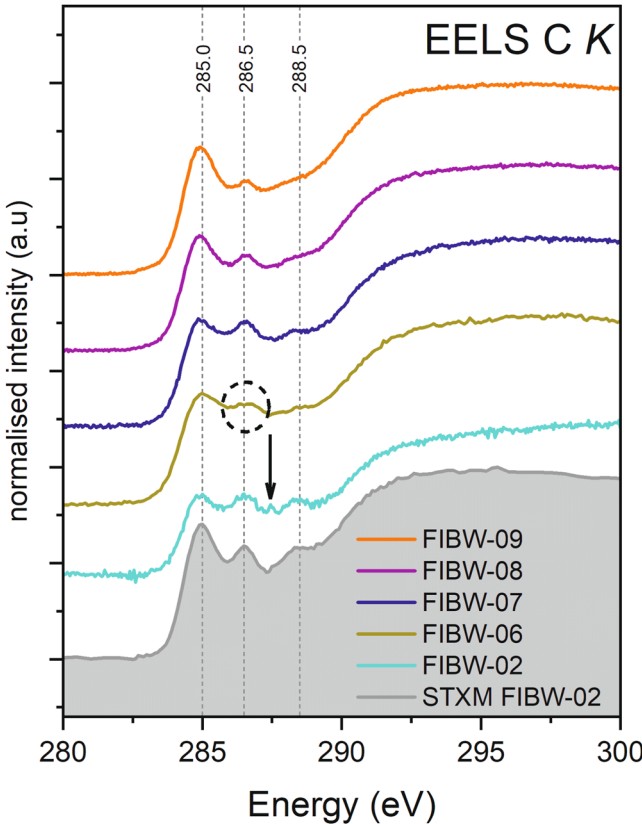

**Fig. 6 | Monochromated EELS data at the carbon K-edge.** The three main absorption bands of IOM are marked in comparison with STXM data. The additional aliphatics band in FIBW-02 is marked by an arrow, whereas the splitting of the 286.6 eV band due to both carbonyl and nitrile bonding is encircled.

**Fig. 5 | Quantification of the carbon K-edge STXM data.** Values have been quantified according to the procedures of Vinogradoff et al. (2017)[13] with comparison to Paris insoluble organic matter (IOM) and Murchison IOM data. Error bars are 1σ. See text for details.

environments may also relate to certain morphological characteristics and may vary on a nanometer scale within a given sample. One fingerprint detected in extracted IOM has been dubbed "IOM-like", which shows three main absorption peaks at ~285 eV (the onset varies by a few 0.1 eV depending on the technique and on the sizes of the poly-aromatic domains), ~286.5 eV, and ~288.5 eV as fine structure on the general π* and σ* broad absorption features. Sometimes, a fourth peak at around 287-288 eV is detected in certain types of organic matter due to the presence of more labile aliphatic C-C chains that interconnect the polyaromatic domains. A second class of organic matter was tagged "aromatic" or "highly aromatic", where the C K-edge onset at ~285 eV dominates the general absorption edge over all other minor bands. Both the "IOM-like" and "aromatic" signatures are also found in OM in the recently returned samples from asteroid Ryugu[37]. However, a highly aliphatic C K-edge functional chemistry has been observed in different work on Ryugu samples by STXM[36], which shows more similarities to cometary OM analyzed in IDPs[38]. A further C K-edge signature was reported in Ryugu OM and certain carbonaceous chondrites[14,15,37], where the aromatic 285 eV band is diminished and the spectra are dominated by the 288.5 eV (carboxylic) and 290.3 eV (carbonate) bands. This signature is more likely associated with fine-grained, diffuse OM and indicates advanced fluid interaction.

The Winchcombe samples analyzed here show some general C K-edge features in EELS and STXM spectra. All extracted lamellae display a rather homogeneous functional chemistry at the C K-edge that does not vary strongly between different lamellae or even within the same lamella (Figs. 4 and 6, Figs. S2–6). The two bands at ~285 eV and ~286.5 eV generally characterize Winchcombe OM, whereas in contrast to the typical "IOM-like" signature, the third carboxylic band is much reduced in almost all analyzed spectra. Furthermore, we do not

observe a strong "aromaticity" in most analyzed spectra, which concurs with the generally low abundance of globular organics in all extracted lamellae with only one nanoglobule identified in lamella FIBW-05 (Fig. 2c) and a multiglobular assemblage in FIBW-04, which is also confirmed by the relatively high aromaticity content of this lamella (Fig. 4b). The occurrence of aromatic and globular OM is usually an indication of less severe aqueous overprint, because in highly aqueously altered meteorites such as petrologic type 1 carbonaceous chondrites, nanoglobules or globular OM are rare[2]. We therefore explain the scarcity of globular and highly aromatic ("aromaticity" > 1.0) OM within Winchcombe by a slightly more advanced parent-body alteration within the lithologies (CM 2.3/2.4) from where these organic grains were extracted when compared to Paris or Murchison. It therefore demonstrates that nanoglobules and multiglobular OM more likely survive within minimally aqueously altered samples such as CR chondrites or in minimally altered CM chondrites such as Paris, Murchison, or Maribo[2,12,13].

The fact that we do observe a general absence or at least very low abundance of the COOH-feature at 288.5 eV in all of our STXM and EEL spectra with both techniques on different length scales confirms that this is an indigenous feature of the Winchcombe sample and not due to some analytical artifact. Previous work has shown that the appearance or absence of this feature can be highly heterogeneous even in less aqueously altered samples than Winchcombe, such as Murchison, Paris, Maribo, or Renazzo (CR)[12–15]. This indicates that the carboxylic functional group relates to a more labile component of the meteoritic OM, such as carboxylic and amino acids, that is easily re-distributed by fluid reactions. Indeed, analyses of amino acids usually distinguish between "free" amino acids and those bound in larger complex molecules, so-called amino acid precursors that have to be released by "acid hydrolysis" for further analysis[31]. Candidate molecules of these amino acid precursors could be compounds with higher molecular weights containing amide bonds[32]. This implies that the carboxylic C K-edge fingerprint occurs in both pristine OM as part of the "IOM-like" chemistry as well as in OM in more altered lithologies or Ryugu OM together with carbonates[14,37]. Further findings suggest that its abundance increases with advancing alteration[13–15].

We therefore draw two main conclusions from these observations: the carboxylic C K-edge fingerprint certainly relates to a labile, i.e., "SOM-like" chemical characteristic of OM that can be easily extracted by solvents. Some pristine acidic components such as amino acids or their precursors are likely associated with the appearance of this feature in highly primitive OM in less-altered samples such as CR chondrites. However, upon proceeding parent-body alteration, such carboxylic components disappear in C K-edge spectra of large organic grains. These small molecules then likely disperse into the matrix as diffuse components and constitute a second, strongly altered fraction of SOM in a similar way as hydrolyzed amino acids. Therefore, for Winchcombe, we conclude that parent body alteration was advanced enough to redistribute this labile component of the initial OM, which was then no longer an important part of the larger organic grains that we analyzed.

## Organic matter N K-edge functional chemistry

STXM analyses of the nitrogen functional chemistry reveal weak, albeit distinct, absorption bands at 398.8 eV and 399.8 eV due to C-N double (imine, i.e., pyridinic N where N substitutes for a C in a benzene ring) and triple (nitrile like in HCN) bonding (Fig. 7). We observe that the relative intensities of the imine and nitrile bands vary between the samples, but generally, the second (nitrile) band is more prominent. However, nitrile N exhibits a very intense 1s-π* absorption band, such that its high intensity must not necessarily reflect high concentrations relative to the other N-bearing species[39]. Further fine structure could not be unambiguously identified in the spectra due to low

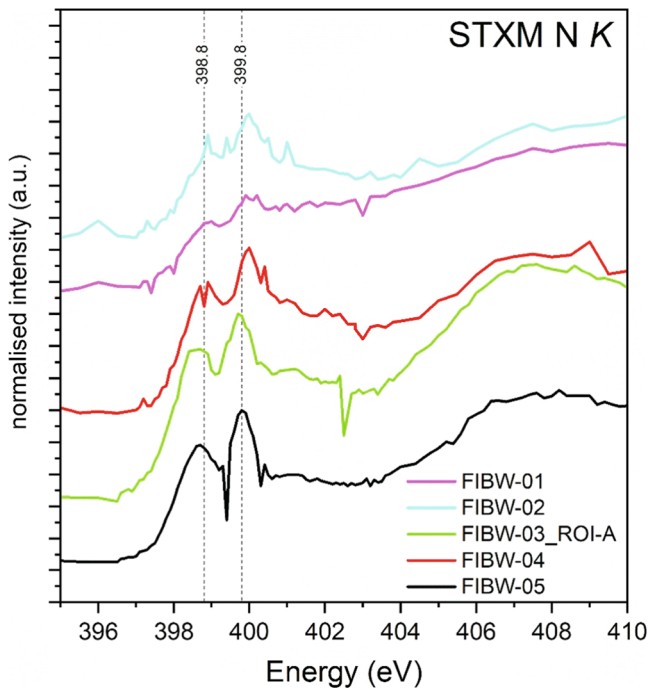

**Fig. 7 | STXM data at the nitrogen K-edge.** The two main absorption bands due to imine and nitrile bonding are marked.

signal-to-noise data above the main absorption edge (~400 eV), a typical limitation of STXM studies at the N K-edge.

The use of a low-noise direct electron detector allowed observations at the N K-edge by EELS that are much more complex and diverse than in STXM data (Fig. 8). These bands (summarized in Table S1) are clearly visible in the spectra in spite of relatively low signal, especially after applying denoising with statistical methods such as principal component analysis (PCA) (Figs. S7 and S8)[40], to which data acquired on direct or hybrid pixel detectors are particularly well suited due to the near perfect Poisson characteristics of the remaining noise present[41]. Hence, in addition to the two bands described above, further fine structure above 400 eV can be resolved. The distinct absorption band below 402 eV at 401.1–401.3 eV in our spectra could be suggestive of the presence of the amino acid L-alanine, which absorbs at 401.1 eV[42]. Furthermore, this amino acid also absorbs at 405.9 eV[42,43], and the tentative detection of a band at 405.7–405.9 eV in some of our spectra could support the presence of this amino acid (Fig. 8). Generally, bands around the absorption energy ~405.7–405.9 eV hint at the concurring presence of different amino acids, as many amino acids in addition to L-alanine, such as L-threonine or L-glutamine, absorb in that energy range[42,43]. Measurement of bands in this energy range therefore suggest the "in-situ" detection of amino acids, that is, within a minimally processed extraterrestrial sample. In addition, advanced statistical analysis using Blind Source Separation (BSS)[44] algorithms suggests nanoscale variations of the N K-edge fine structure within the OM, especially for those spectral components with features at energies above 400 eV (Fig. S9 and associated discussion). The band at 405.9 eV assigned to L-alanine is particularly prominent in one of the two spectral components identified by BSS, providing further confidence about the identification of this band.

Additional N K-edge fine structure above 402 eV is observed in different regions of the lamellae. These bands can be explained by the presence of amine (C-NH$_x$- at ~402.2 eV) and amide (CO-NR$_2$ at ~402.4 eV) functional groups with varying hydrogen substitutions[19]. Furthermore, bands at around 402 eV can also be related to the presence of aromatic N-heterocycles. Extended fine structure in our analyzed N K-edge EEL spectra (Fig. 8) indicates further absorption at

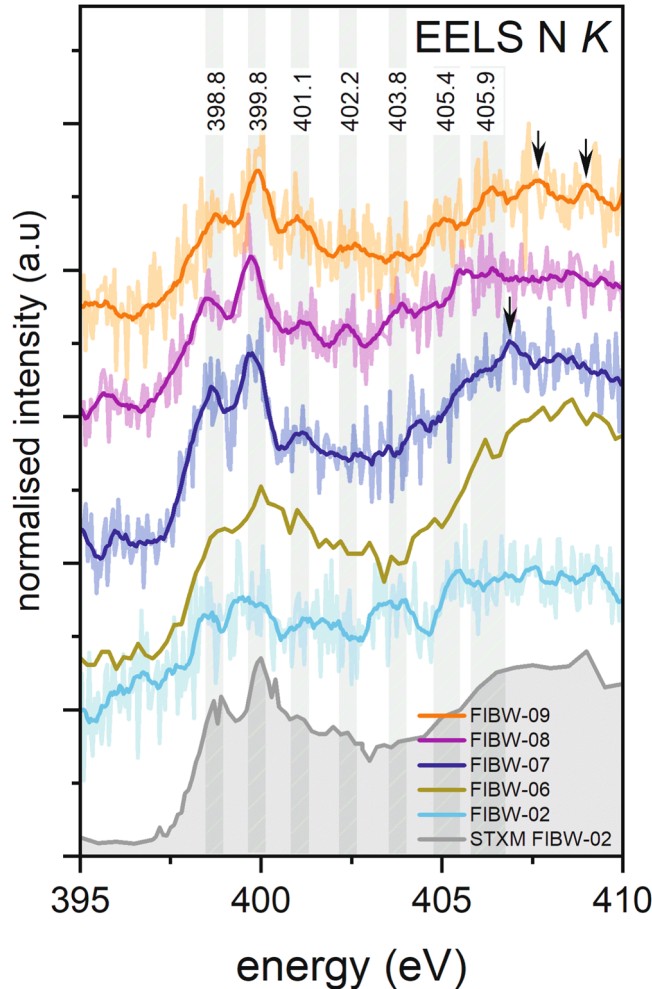

**Fig. 8 | Monochromated EELS data at the nitrogen K-edge compared with STXM data.** The main absorption bands discussed in the text are indicated. Note: the spectrum for FIBW-06 was acquired with a poorer energy resolution of 0.3 eV than all other N-K EELS data presented. It is included here for completeness and comparison with all acquired STXM data. See text for details.

## Formation pathways of nitrogen-bearing compounds

The nitrogen functional chemistry of the OM in our samples summarized above is of critical importance to interpret organic matter formation pathways. There is in general much less information available about N K-edge fingerprints in extraterrestrial OM compared with the C K-edge. This is mainly because N/C ratios in most types of meteoritic and cometary OM are <0.05 and low signal-to-noise data hamper definite conclusions about bonding environments. Only in certain types of N-enriched samples in ultra-carbonaceous Antarctic micrometeorites[45] or small regions in carbonaceous chondrites such as NWA 852 (CR)[46] or Renazzo[15], do higher N/C ratios allow qualitative statements about the functional chemistry. Even then, spectra easily degrade by continuous synchrotron radiation or electron exposure[47], which makes analyses at the N K-edge quite challenging. Furthermore, detector efficiencies to date have not generally been suitable to analyze high quality N K-edge spectra.

The C-N double (imine) and C-N triple (nitrile) bonding environments in both our STXM and EEL spectra could represent a pristine component of OM and might even contain observed N-isotopic anomalies, both in hotspots and coldspots[10,16,48]. This signature was probably inherited from circumstellar or cometary HCN molecules that are observed in comets or the interstellar medium (ISM) spectroscopically[49–51]. These molecules are also important precursors to the formation of amino acids by the Strecker-cyanohydrin synthesis pathways that involve aldehydes and ammonia[52]. Unsaturated nitrile bonds also play a vital role in the formation of β-amino acids by Michael addition[31]. However, the double and triple bonding environments are highly susceptible to alteration and modification[19], which is why their detection here points at a relatively pristine state of the OM. We therefore conclude that the overall low abundance of N K-edge bonding environments might be a reason why amino acid concentrations in Winchcombe are similarly low as well[26]. However, acknowledging that extensive parent body alteration would have led to its oxidation and hydration, we qualitatively conclude that Winchcombe still preserves pristine nitrogen functionality in its organic matter.

We also observe extended fine structure above 400 eV. The tentative "in-situ" (that is, within the preserved petrographic context) detection of different amino acids such as L-alanine in our N K-edge EEL spectra is important. This extraterrestrial amino acid has also been detected in Winchcombe SOM[26] as well as in Ryugu OM[53]. Furthermore, it has been previously identified in other types of carbonaceous chondrites[4,42]. The fact that we likely observe this amino acid within a minimally processed sample suggests that its occurrence in other work on processed extracted samples is not due to hydrolysis from amino acid precursors during the extraction process, but is an indigenous feature of the OM.

Further fine structure around 402–403 eV is likely due to the presence of different amine and amide functional chemistry within amino acids and their precursors. The abundances of these bands vary between different lamellae and even within regions of the same lamella (Fig. 8). This is in contrast to the two bands below 400 eV that occur in all spectra, regardless of selected regions. These functional groups have also been found in less aqueously altered CR chondrites such as Renazzo[15]. The varying intensities of these bands on a sub-μm scale within a FIB lamella indicate fluid processes and loss of H atoms in microchemical environments that transform primary to tertiary amine and finally to amide groups within different types of amino acids.

However, the overall strength of the absorption bands above 400 eV is generally very low. This could imply that HCN-related chemistry dominates the Winchcombe OM, although, as explained above, high nitrile absorption may not necessarily reflect high concentrations[39], making quantification of these functional groups in a similar manner as for C K-edge difficult. We could still argue that the formation of amino acids and any related amine and amide functional

around 406.9 eV, 407.6 eV, and 408.9 eV, which supports the presence of different types of N-heterocycles and nucleobases (marked with arrows in Fig. 8) such as imidazole, pyrimidine and adenine[42]. There is only limited fine structure at around 403.8 eV, which points to the presence of nitro (NO₂-) groups, and some absorption at 405.4 eV due to nitrate bonding that both indicate oxidation and alteration of OM[1,19]. Only in lamellae FIBW-02, FIBW-08, and FIBW-09, is there some indication of nitro and nitrate bonding environments. Lamella FIBW-02 also contains OM with indications of carboxylic bonding (Fig. 4a). Therefore, in general, oxidation of the Winchcombe OM appears to be minimal, which is also evident to some degree by the absence of carbonate in the C K-edge data. A very weak carbonate feature at ~290 eV is only observed in one STXM spectrum (FIBW-04). In general, the abundance of all bands related to N-H (amine/amide) chemistry and N-heterocycles are generally much weaker than the strong bands of imine and nitrile below 400 eV. Finally, it should be noted that meteoritic OM is a highly complex material. Consequently, positions and shapes of identified bands may not agree perfectly with assignments from reference papers[42], which may have been acquired on pure substances in different experimental conditions. Therefore, band positions of our N-containing compounds are likely spread over a finite energy range and not restricted to exact values from the literature.

chemistry was probably hampered by low abundances of the ammonia-rich ices necessary to synthesize different types of amino acids on the CM parent body. In either case, our results qualitatively agree with the conclusions drawn by Chan et al. (2022)[26], who observed a low amino acid abundance in Winchcombe by SOM analyses indicating advanced aqueous alteration. However, in contrast, they also reported high free:total amino acid and nonalkylated:alkylated PAH ratios which seem to suggest little aqueous overprint. The limited aqueous alteration extent is also documented in our work by the weak absorptions at around 403.8 eV and 405.4 eV due to oxidized nitro and nitrate bonding, respectively. We therefore argue that the appearance of these bands in meteoritic OM in general could be due to terrestrial overprint and oxidation rather than parent body alteration.

Bands at around 402 eV and above at around 406.9 eV, 407.6 eV, and 408.9 eV likely relate to the presence of aromatic N-heterocycles (Fig. 8, marked with arrows). Aromatic nitrogen-containing heterocycles such as nucleobases are also detected in meteoritic OM[54], but are usually at very low abundances. N-heterocycles such as pyrrole ($C_4H_4NH$) and imidazole ($C_3H_4N_2$), but also nucleobases such as guanine and adenine, have been detected in CM chondrites like Murchison[4,30]. Important N-containing polyaromatic molecules were also found by infrared spectroscopy in the ISM[55]. Other work has demonstrated the occurrence of the one-ring ("pyrimidines") compound uracil and the two-ring ("purines") compound xanthine as extraterrestrial nucleobases in the Murchison meteorite[30] and in Ryugu OM[56]. As mentioned above, we do see an indication of these bands at higher absorption energies with varying intensity despite the low signal. Furthermore, the presence of C-N double bonding environments, i.e., imine functional groups, could also point to the presence of N-heterocycles. Pyrrole also absorbs at around 403.7 eV[42], which overlaps with the aforementioned nitro groups, and the observed band could therefore be assigned to this N-heterocycle as well. One important N-heterocycle present could be specifically imidazole[57], which is known to serve as a possible catalyst to synthesize other organic monomers like nucleotides and amino acids and might play an important role on the chemical evolution on asteroids[29]. Its alkylated homologs were identified in a diverse suite of meteorite samples such as Murchison, Tagish Lake, and Murray (CM). Imidazole also absorbs in the same region as nitrile at ~400 eV[43], so the presence of nitrile might be questionable[19]. However, at our high energy resolution, C K-edge EEL spectra clearly show the presence of two peaks at around 286 eV that are only a few 0.1 eV apart (encircled in Fig. 6) that can therefore be assigned to both ketone/aldehyde/carbonyl and nitrile bonding. The strong absorption at around 400 eV could therefore be related to the presence of both nitrile bonding and imidazole molecules. We therefore conclude that these important prebiotic molecules are present in analyzed Winchcombe OM, though future higher resolution work on OM could take these observations further.

The work presented here demonstrates that high-spatial resolution spectroscopy techniques with superior energy resolution and high-end low-noise direct electron detectors facilitate the investigation of important biologically relevant molecules, such as amino acids and N-heterocycles, within a minimally processed meteorite sample. Specifically, STEM-EELS investigations with high spatial resolution are perfectly suited to identify local differences in complex bonding environments of organic matter, while advanced statistical analysis highlights the potential of mapping of such features at the nanoscale[44]. Our investigations show that these N-bearing compounds are part of the larger organic grains in extraterrestrial samples. It also supports the importance of analyzing samples with minimal terrestrial overprint such as the Winchcombe meteorite that provides a unique sample of the early solar system for organic analyses. This type of investigations and the combination of complementary techniques deployed here will be directly applicable to the possible detection and analysis of prebiotic molecules in the recently returned OSIRIS-REx samples from asteroid Bennu.

## Methods

### Samples

We investigated polished section P30544 (BM.2022, M2-46) of the Winchcombe meteorite, with a size of about 2.5 mm², curated by the Natural History Museum (NHM), London, UK[21]. This fragment contains three lithologies of the Winchcombe meteorite according to the detailed petrographic work of Suttle et al. (2023)[22] (Fig. S1). Lithology B is heavily altered (petrologic type 2.1) and consists of abundant tochilinite-cronstedtite intergrowths (TCIs) with very low metal abundances. We did not extract any OM lamella from this lithology, because distinct organic grains could not be recognized within this unit, although diffuse organic material occurs (Fig. 1b). The second, less altered lithology (lithology H in Suttle et al. 2023) is also present within this section (petrologic type 2.3/2.4) and is characterized by a highly variable alteration extent, where between 30% and 80% of the initial anhydrous silicates have been replaced[22]. This lithology is dominated by type II TCI clusters with distinct lamination textures. We detected organic grains within this lithology and extracted three lamellae (FIBW-06, FIBW-07, FIBW-08) from this unit. More than half of the thin section (~1.4 mm²), however, represents the so-called "cataclastic"[22] or "clastic"[58] matrix separating the other, more distinct lithologies from one another. This matrix has a highly heterogeneous texture and variable alteration extents, with abundant subangular fragments of different CM chondrite components. We extracted six lamellae (FIBW-01 – FIBW-05, FIBW-09) from this matrix in different areas of the thin section (Fig. S1). Areas from which these lamellae were extracted are characterized by abundant phyllosilicates and TCI phases as well as heterogeneously distributed sulfide grains with subangular morphologies. Before subsequent synchrotron experiments, all lamellae were documented by low kV (≤5 kV) Scanning Electron Microscopy (SEM) for morphological characterizations. We did not perform STXM and EELS experiments on all lamellae, because a number were either too thick for subsequent EELS analyses or suffered damage during transport between the Diamond Light Source (DLS) and SuperSTEM laboratories. Some additional lamellae were prepared specifically for EELS experiments after the STXM beamtime (FIBW-07 - FIBW-09), so we do not have STXM data of those later lamellae.

### Scanning electron microscopy (SEM)/Focused ion beam (FIB)

The general petrography of the different lithologies were documented using a Zeiss EVO LS15 SEM (20 kV) at the NHM London. Organic matter was then identified using an FEI Quanta 650 field emission SEM (6 kV) at the NHM and a Hitachi Ethos NX5000 FIB-SEM (3 kV) at the SuperSTEM laboratory. The OM can be clearly identified by its darker contrast compared to the surrounding matrix. Electron-transparent lamellae were prepared with the Hitachi instrument using standard FIB preparation protocols for the fabrication of samples intended for low-kV STEM observations[59]. Lamellae were lifted out to Hitachi NanoMesh support grids and thinned using successively lower currents of 30 kV $Ga^+$ to a thickness of a few hundred nm. Smaller areas of interest were further thinned using currents and energies down to 5 kV and 20 pA $Ga^+$. Areas around these windows were kept thicker as a natural frame for stability and heat dissipation[60]. Finally, 1 kV $Ar^+$ polishing was applied directly within the FIB-SEM instrument using a dedicated 'third beam' $Ar^+$ source, using 15 s exposures at 4 different incident directions for up to 10 min in total to remove remnant damage from low-kV Ga milling. Although $Ar^+$ ions can also induce damage to thin samples, especially at higher energies, broad-beam low-energy $Ar^+$ polishing has been widely shown to be highly effective at removing most surface amorphization and Ga-implantation-related damage following FIB lamella preparation[61-63]. The application of $Ar^+$ polishing directly in the FIB offers here an additional degree of control and precision in

targeting during this final 'cleaning' procedure, so as to avoid redeposition of support grid material onto the lamella surface, and thus furthering lower any risk of damage[64]. SEM exposure of the thin areas was intentionally limited to mitigate any electron beam induced effects: when practical, there was no further exposure to the electron beam following the last Ar⁺ polishing 'run' to ensure as unexposed a surface as possible for spectroscopy observations. For lamellae targeted at organic grains deep below the surface of the polished section, we refrained from protecting the area of interest with a Pt strap or coating overlayer to avoid contamination with extraneous material. For lamellae aiming at surface visible OM, a Pt strap was necessary, but the lamella was then overturned for milling to reduce any risk of redeposition onto the OM as it was thinned to electron transparency (<30 nm in the thinnest regions, and generally <~100 nm overall for 60 kV investigations). Given the very sensitive nature of the organics studied here, it is impossible to fully exclude the possibility of some sample alteration during ion beam thinning. This may be minimized in the future through the further exploration of the effect of low energy Ar⁺ milling on organics, and the use of cryogenic temperatures for Ga⁺ milling. Nevertheless, the clear fine structure observed across all the specimens studied here, with sharp, unbroadened spectral bands close to the instrumental energy resolution[65], and irrespective of possible, inevitable small variations in preparation parameters between lamellae, suggests that damage, if present, was minimal.

### Aberration-corrected scanning TEM (UltraSTEM)

Electron-transparent lamellae were investigated on a dedicated aberration-corrected monochromated Nion UltraSTEM100MC−Hermes operated at 60 kV in so-called "gentle" STEM conditions (low acceleration voltage to avoid knock-on damage to carbon-based material, ultra-high vacuum conditions to prevent chemical etching of the sample, or surface contamination from adventitious carbon). The instrument is equipped with a cold field emission electron source with a nominal energy spread of around 0.3 eV (as measured by the full-width at half-maximum, FWHM of the zero-loss peak, ZLP). The microscope features an ultra-stable stage, conventional bright field (BF, 0–6.5 mrad angular range in the conditions used for imaging) and high-angle annular dark field (HAADF, 90–190 mrad angular range) imaging detectors, as well as a Nion IRIS high energy resolution energy loss spectrometer equipped with a Dectris ELA hybrid-pixel direct electron detector optimized for Electron Energy Loss Spectroscopy (EELS) at low acceleration voltages. The use of direct electron detection provides major advantages, such as low noise (any remaining noise has near-Poisson characteristics) and detection quantum efficiency, even in low-dose conditions[41]. The probe forming optics were adjusted to provide a 0.1 nm probe with a beam convergence of 30 mrad (half-angle), while a collection half-angle of 44 mrad was chosen for EELS analysis. The energy resolution for the C and N K-edge EELS measurements was in a range of 50−90 meV, depending on the position of the monochromator slit, adjusted as a compromise between energy resolution and probe current/signal levels. The resulting beam current was ~3−5 pA.

EELS spectrum images (SI) were acquired by rastering the electron probe across an area of the sample and acquiring an EEL spectrum at each point. Averaged spectra from regions of interest were energy-calibrated with respect to the exact position of the ZLP and the energy offset compared to spectra from a graphite standard sample, acquired under identical conditions. Possible beam damage was monitored through careful examination of the spectral fine structure before, during, and after SI acquisition.

The exact electron dose on the sample can vary from dataset to dataset, as it is dependent on the size of the scanned area and the duration of the spectrum image acquisition. Nevertheless, the order of magnitude of the electron dose remains similar across all datasets. The 3-5 pA probe current was delivered to the sample in a probe of 0.1 nm diameter in size and rastered across large regions of the sample over 100 nm in length, in a sub-sampled way (that is to say, the distance between pixels in a dataset was larger than the probe size; we used typically 1 nm pixels or larger)−which is one of the experimental procedures adopted to mitigate possible beam damage (so that if damage does occur at a given probe position, the neighbouring pixel is far enough away that delocalized damage to the sample is limited). For a typical dataset, the electron dose per spectrum image is $3-5 \times 10^4$ e⁻/Å².

EELS SIs were denoised using Principal Component Analysis (Fig. S7)[40]. EELS N K-edge spectra are overlaid against Savitzky-Golay (2nd order polynomial, 20-point window) smoothed lines as a guide to the eye (see supplementary information for more details, including spectra plotted without any additional smoothing, Fig. S8). The spectra were background subtracted using a standard decaying power-law model.

### Scanning transmission X-ray microscopy (STXM)

Scanning transmission X-ray microscopy (STXM) and X-ray absorption near-edge structure (XANES) measurements were performed at the I08 beamline of Diamond Light Source (DLS), UK. Spatially correlated energy dependent image stacks (~5 × 3 μm, 100 × 60 pixels) were acquired at the carbon (278–310 eV) and nitrogen (390–420 eV) K-edges with a nominal beam size of ~40 nm and a dwell time per pixel of 10 ms. For both the carbon and nitrogen measurements, the step size was 0.1 eV over the main edge and where important diagnostic features occur, and between 0.2 and 0.5 eV across the rest of the scan. The MANTiS program[66] was used to align the image stacks, normalize to the background X-ray intensity ($I_O$), and finally extract XANES spectra from every pixel in the full FIB lamella and specific regions of interest (ROIs). Spectral fitting was performed using the Athena software package[67] utilizing the methodologies of Le Guillou et al. (2014)[17] and Vinogradoff et al. (2017)[13].

## Data availability

The STXM and EELS processed data generated in this study have been deposited in the research data repository of the University of York under accession code https://doi.org/10.15124/5e16dd60-d078-43cd-98fb-5db41e042547. Further images and processed EELS/STXM data generated in this study are provided in the Supplementary Information file. Further correspondence and requests for materials such as raw images as TIFF should be addressed to C.V.

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

## Acknowledgements

C.V. and J.L. acknowledge support by the DFG through SPP1833 grants (VO1816/3-1 and LE 3279/2-1 & 2-2). A.J.K. and P.F.S. were supported by UKRI (MR/T020261/1) and the STFC, UK (ST/V000799/1). SuperSTEM is the U.K. National Research Facility for Advanced Electron Microscopy, supported by the Engineering and Physical Sciences Research Council (EPSRC, UK) via grant numbers EP/W021080/1 and EP/V036432/1. We thank Burkhard Kaulich and Majid Kazemian for help with STXM analyses and acknowledge Diamond Light Source for time on beamline I08 under proposals MG30183 and MG31026.

## Author contributions

C.V., J.L., and A.J.K. designed research; C.V., D.K., J.L., A.B.M., K.E.H., A.J.K., P.F.S., C.L.B., T.A., and Q.M.R. performed research; C.V., D.K., Q.M.R., A.B.M., A.J.K., P.F.S., C.L.B., and T.A. analyzed data; all authors participated in interpretation, C.V. wrote the paper, all authors edited the paper and have approved the submitted version.

## Funding

## Competing interests

The authors declare no competing interests.
