## [Peer Review File NEW · Nature Communications]

High-spatial resolution functional chemistry of nitrogen compounds in the observed UK meteorite fall WinchcombeREVIEWER COMMENTS

Reviewer #1 (Remarks to the Author):

The manuscript "High-Spatial Resolution Functional Chemistry of Nitrogen Compounds in the Observed UK Meteorite Fall Winchcombe" presents new data combining monochromated electron energy loss spectroscopy with a scanning transmission electron microscope and synchrotron x-ray absorption near edges spectroscopy (XANES) to argue for the detection of N-rich organic compounds, such as amino acids, with a possible nebular or molecular cloud origin. While the data are valuable in expanding the available information on organics from fresh fall carbonaceous chondrite meteorites, the interpretation is incomplete and not fully supported by the data analysis.

The claim that measurements represent the first high-resolution detection of N-rich organic compounds in an unprocessed meteorite sample is incorrect; the results are not the first demonstrating N-rich compounds at high resolutions, and the sample was FIB processed. Alexander et al. (<https://doi.org/10.1016/j.chemer.2017.01.007>) Figure 3, shows data combining C and N XANES with TEM and NanoSIMS to demonstrate the presence of nitriles in a CR2 meteorite. In both the submitted manuscript and the Alexander et al. paper, the samples were prepared as FIB lift-out lamella. The FIB preparation does preserve the petrographic context, but it is not unprocessed. The use of "unprocessed" was perhaps intended to mean without exposure to solvents involved in the isolation of the organic components from the meteorite minerals. However, it would more accurate to say FIB-extracted, rather than unprocessed. This accurately conveys the trade-offs in terms of preservation of soluble components and petrographic context, and possibility of FIB-induced damage.

The manuscript asserts that the monochromated STEM-EELS data can be used to map the concentration of amino acids and nucleobases in meteoritic OM, but does not provide convincing evidence of unique detection of amino acids or nucleobases. The extracted N K-edge EELS spectra (Fig. 7) show convincing evidence for immine and nitrile groups, similar to those in Alexander et al. 2007 (Fig. 3). The identification of spectral features above 400 eV assigned to amino and other groups are highly speculative, and may in fact be artifacts of the smoothing window chosen for the Savitzky-Golay spectral processing. Data from compound-specific mixtures to establish detection limits and preservation through exposure to the x-ray, ion and electron beam exposures are needed to provide adequate confidence in the detection of amino acids and nucleobases. The soluble organic compounds are at much lower abundance, and are much more susceptible to damage than the insoluble compounds, and thus the reported detection is not convincing. Better presentation of the data processing, e.g., exactly what PCA and smoothing parameters were used, and use of analog standards for validation could substantially improve the manuscript.

Overall, there is important value in the measurements, but primarily to the meteoritics and planetary science community.

Reviewer #2 (Remarks to the Author):

[1] Overview comments

At first, I would like to thank the opportunity to review the manuscript "High-Spatial Resolution Functional Chemistry of Nitrogen Compounds in the Observed UK Meteorite Fall Winchcombe" by Vollmer and co-workers. Various analyses of this meteorite are underway, led by a research group in the United Kingdom. It is of high scientific descriptive value because of its rapid and reliable analysis for light elements and prebiotic organics while minimizing exposure to the Earth's atmosphere. Among these, the authors have investigated non-destructive spectroscopic measurements with microscopic observation in order to decipher the characteristics of carbonaceous meteorites from their organic-specific spectra. In particular, the careful spectroscopic measurement of the nitrogen band and the highly accurate data describing its distribution is a unique study. It is expected that the following main points should be fitted to the latest knowledge. In other words, it is meant to be a complementary opinion to make the value of Winchcombe and those new findings robust.

[2] Lines 2-4, 18-25 : Perhaps there was a slight gap between the timing of the submission of this paper and the publication of the findings discussed below. The authors mention at the beginning of the abstract that they have identified amino acids from Winchcombe meteorite. Amino acids are compounds with amino and carboxyl groups (refs. 44, 45). The presence of their individual polar functional groups is frequently observed in various spectroscopies. On this context, the complementary evaluation with the previously reported identification of amino acid molecules by destructive analysis is also convincing. The water content of the meteorite would be one reference for estimating the history of aqueous alteration (ref. 43). In this sense, comparative feedbacks with Bates et al. 2023 & Daly et al. could be discussed complementary.

-Bates, H.C., King, A.J., Shirley, K.S. et al. (2023) The bulk mineralogy, elemental composition, and water content of the Winchcombe CM chondrite fall. Meteoritics & Planetary Science, doi: 10.1111/maps.14043.

-Daly, L., Suttle, M. D., Lee, M. R. et al. (2023) Nano-Scale Heterogeneity in the Extent of Aqueous Alteration within the Lithologies of the Winchcombe CM Carbonaceous Chondrite. Meteoritics & Planetary Science.

[3] Lines 21-27: Based on the non-destructive measurement, the authors discuss indigenous SOM (refs. 44, 45) and IOM profiles (e.g., STXM data interpretation) throughout the paper. In particular, I would add the need for verification of consistency with the latest findings on IOM to feedback on the present data.

-Sephton, M.A., Chan, Q.H., Watson, J.S. et al. (2023) Insoluble macromolecular organic matter in the Winchcombe meteorite. Meteoritics & Planetary Science, doi: 10.1111/maps.13952.

[4] Lines 37-48: I think that that the early view literature describing the process of survey, sampling, initial evaluation, and curation immediately after the Winchcomb meteorite fall is an important paper in terms of assuring the value of the sample.

-Russell, S.S., King, A.J., Bates, H.C. et al. (2023) Recovery and curation of the Winchcombe (CM2) meteorite. Meteoritics & Planetary Science, doi: 10.1111/maps.13956.

[5] The authors mention nitrogen (N)-containing heterocyclic compounds in samples returned from carbonaceous meteorites and carbonaceous asteroids (refs. 24-27, 48). Their point of view is very interesting for astrochemistry and prebiotic molecular evolution. On the other hand, however, it may be necessary to provide additional explanation as to whether or not the signal attribution of imino groups (e.g., -C=N-) derived from heterocyclic compounds can be evaluated with any degree of accuracy in the present

spectroscopic measurements. In fact, nitrogen (N) exists in various chemical forms such as ammonia (NH₄⁺), amine (-NH₂), imine (-C=N-), or nitrate (-NO_x) forms. Would it be possible to show an organized (e.g., table data) list of the raw values of the spectral chart? I would like to suggest that the key points of the qualitative attribution of this nitrogen signal be described in the results of the main text or in the SI script.

That's all, thank you.

Reviewer #3 (Remarks to the Author):

Review of the paper "High-spatial resolution functional chemistry of nitrogen compounds in the observed UK meteorite fall Winchcombe" by Vollmer et al. for Nature communications.

This article reports a systematic organics study of several FIB sections from the recently felled UK meteorite Winchcombe. I have no doubt that the Winchcombe is very important and expand for our understanding of the formation and nature of the Solar System. I believe that this work will be of substantial interest to the extraterrestrial and planetary science communities. I think Nature communications is a good target after minor revision.

1. noteworthy results

The authors presented functional chemistries of carbon and nitrogen, and micro-scale textural characterizations of several FIB sections from the Winchcombe meteorite. They showed that the samples have imine and nitrile functions based on analyses by SuperSTEM-EELS and SR-based STXM-XANES at a sub-micrometer scale. This important finding could derive better understanding of evolution of life-related organics in a carbonaceous body. In fact, after a flinging of Uracil (C₄H₄N₂O₂, Oba et al., Nat. Comm, 2023) in Ryugu samples which is one of the four nucleobases in ribonucleic acid, nitrogen-bearing organics in less or uncontaminated extraterrestrial materials draws our attention of astrobiology point of view.

2. Will the work be of significance to the field and related fields? How does it compare to the established literature? If the work is not original, please provide relevant references. The characterization of Winchcombe meteorite sample with minimal terrestrial contamination is extremely important because we can analyze uncontaminated Ryugu samples at the same time and compare each other. These two carbonaceous materials expand our knowledge about extraterrestrial organics and astrobiology. Therefore, this manuscript is highly related with our planetary science community. So, my impression of this manuscript is unique and relevant. All cited references look OK for me.

3. Does the work support the conclusions and claims, or is additional evidence needed? Agree with authors conclusions. Personally, I would like to see hydrogen, carbon and nitrogen isotopic compositions of C-rich area in FIB sections, especially ROI containing imine and nitrile functions. Those isotopes may help to decipher nature, origin and evolutionary history of organics having imine and nitrile functions. If authors have a chance to access to NanoSIMS, please conduct isotope imaging analyses. This improved the manuscript.

4. Are there any flaws in the data analysis, interpretation and conclusions? Do these prohibit publication or require revision?

I think that data analysis, interpretation and conclusions are good. However, I have some questions regarding STXM analysis of the sample. Please see my questions/comments below.

5. Is the methodology sound? Does the work meet the expected standards in your field? Methodology is appropriate, and the original data included in the manuscript are high-quality by a well-designed analytical approach. I would appreciate if authors could describe more details about FIB methodology to the Winchcombe even author cite a proper reference.

6. Is there enough detail provided in the methods for the work to be reproduced? I think that authors described enough information of their analytical approach to observe C and N functional groups by STXM-XANES and EELS data by sub-micrometer resolution SuperSTEM.

I have some questions/comments below. Overall, the manuscript is well written, the quality and presentation of the data is excellent, and the results have important implications. One additional point would be the asteroid Bennu (CM-type by spectroscopic study on board) samples by NASA O-Rex mission. I think that it would be nice to mention possible genetic link to asteroid Bennu even a little bit speculation.

- 1. If I remember correctly, a STXM is not a sensitive to nitrogen. It requires a few wt% to detect N-bonding peaks with proper signal-to-noise ratio.**
- 2. Please see reference Zubavichus et al (2005, Innershell Absorption Spectroscopy of Amino Acids at All Relevant Absorption Edges. J. Phys. Chem. A, 109, 6998-7000) for amino acid study with STXM C and N-XANES. There is a combination of C and N-XANES spectrum in each amino acid.**
- 3. P5, L106: It is better to have a proper reference for "because aliphatic functional groups are easily disturbed by alteration reactions". Alteration reactions of what?**
- 4. P6, L111: Please clarify this statement of "the STXM analyses did not modify the OM of those lamellae". C-XANES of FIBW-02 by STXM and that of EELS look different to me, for example, (1) the ratio between 285.0eV and 286.5eV, and (2) no 286.6eV peak (or just a sholder?) in FIBW-02 by STXM but that in FIBW-02 by EELS (indicating as arrow in Fig. 6). Could this be some modification during electron and/or photon irradiation? Please see reference "Wang et al., 2009, Radiation damage in soft X-ray microscopy. Journal of Electron Spectroscopy and Related Phenomena 170 (2009) 25-36". In fact, from my experience of C-XANES analysis of Ryugu FIB samples, there were obvious modification in 286.6eV peak (aliphatic) by photon irradiation during STXM analysis. I think that it is better to describe photon and electron doses during your analysis with STXM and EELS.**

Date 6.11.2023

Detailed response to the reviewers' points (original comments in *italics*)

Reviewer #1

The claim that measurements represent the first high-resolution detection of N-rich organic compounds in an unprocessed meteorite sample is incorrect; the results are not the first demonstrating N-rich compounds at high resolutions.

We feel this comment incorrectly characterizes the findings and claims of our submitted manuscript. We do not claim at any point that this paper presents the first ever identification of N-rich compounds in a meteorite sample. Rather, we highlight the first identification of **specific** N-containing organic compounds such as amino acids and nucleobases in a minimally processed sample, i.e., in a sample for microscopic observation produced in a way that preserves the petrographic context of the analyzed organic matter and without the use of any harsh acid or solvent treatment. Furthermore, N K EELS data at the energy resolution of our monochromated instrument has not been reported to date by any other similar work on meteoritic organics.

The reviewer refers to Fig. 3 in the Alexander et al. (2017) review. From a technical point of view, we note that Fig. 3 of this paper only shows N-XANES spectra, and that no EELS data is reported therein. Irrespective of the technique used, we argue that this dataset is not directly comparable to ours, because: 1) we are able to identify bands above 400 eV (using monochromated EELS) and 2) we have solely analyzed minimally processed, i.e., FIB-prepared organic grains and not IOM. Specifically, the "vein" curve in Fig. 3e of that paper shows a (rather noisy) N-K XANES edge, which in principle could be compared to our data, as it too was obtained from a FIB-prepared sample. However, it does not show any distinct absorption feature above 400 eV, which is the partial focus of our analysis. It is true that the N-XANES of extracted IOM in this figure shows some features above 400 eV that might also be assigned to other N-bearing compounds. But not only is this not discussed explicitly by Alexander et al. in their review, this is also in a "heavily processed", i.e., acid-treated IOM residue.

Once more, we never questioned the prior detection of N-compounds in processed samples, where concentrations are enriched. Similarly, the imine and nitrile bands below 400 eV have

been detected before by several authors with similar techniques, and we never claimed to be the first to identify those bands below 400 eV. However, we argue that our detection of bands above 400 eV is the first by monochromated EELS (thanks, as we describe, to the use of direct-electron detectors) in a “minimally processed” sample, i.e., without the use of any acid or solvent treatment.

In both the submitted manuscript and the Alexander et al. paper, the samples were prepared as FIB lift-out lamella. The FIB preparation does preserve the petrographic context, but it is not unprocessed. The use of “unprocessed” was perhaps intended to mean without exposure to solvents involved in the isolation of the organic components from the meteorite minerals. However, it would more accurate to say FIB-extracted, rather than unprocessed.

This is certainly true: no sample is “unprocessed” in a strict sense. We have changed this term to “minimally processed” throughout the manuscript, and we have added a note in the introduction text to clarify what we define as “minimally processed” and why it matters.

The manuscript asserts that the monochromated STEM-EELS data can be used to map the concentration of amino acids and nucleobases in meteoritic OM, but does not provide convincing evidence of unique detection of amino acids or nucleobases. The extracted N K-edge EELS spectra (Fig. 7) show convincing evidence for imine and nitrile groups, similar to those in Alexander et al. 2007 (Fig. 3).

We respectfully disagree with this comment. On the one hand, we point out that we only claim to report on the “identification” of these N-containing compounds on a qualitative basis, not on “mapping” their concentration. Concentrations are indeed low, and we felt the quantification from N-K edge EELS data was too noisy to be reliable, even if it agreed qualitatively with the quantification procedures for carbon functional groups using the carbon K edge data that we do report in the manuscript.

As stated above, the identification of imine and nitrile functional groups is not new to this paper and has been achieved before, albeit mostly in highly processed IOM samples. Nevertheless, we agree with the reviewer that the identification of amino acids and nucleobases in this minimally altered sample is based on noisy spectra. Nevertheless, we feel that the interpretation is reliable and that the detection, however subtle it may be, would not have been possible through other techniques and thus is worth reporting in this paper. Please see our comments below about the processing routines we have applied to the spectra and why we believe that comparing the signal in both raw and processed data demonstrate a convincing identification. In particular, additional processing of the data using machine learning and statistical analysis does point to spatially resolved variation across the observed fields of view of the fine structure associated with the L-Alanine band, thus suggesting that nm-scale mapping of functional chemistry in meteoritic samples may be possible using our combination of techniques.

The identification of spectral features above 400 eV assigned to amino and other groups are highly speculative, and may in fact be artifacts of the smoothing window chosen for the Savitzky-Golay spectral processing.

In order to clarify any concerns regarding the N K EELS data interpretation we have introduced a section on EELS data processing in the Supplementary Information file. We note that the bands in the N K spectra were evaluated and interpreted from in the PCA denoised data only, and the Savitzky Golay smoothed lines are predominantly added as a guide to the reader’s eye, as due to space restrictions in the final plot we felt the features

were not as obvious as when inspecting the data on a large screen (this is explicitly mentioned in the Methods section).

In the new Figure S8 we plot the very same N K spectra using different levels of spectral smoothing. We would argue that the observed spectral features, here marked with grey bands for reading convenience, are quite distinguishable above the noise level and distinct, even with no line smoothing. In fact, the level of noise of the peaks at higher energies (e.g. 405.9 and 406.9 eV) is similar to that of the π^* bands at 398.8 and 399.8 eV, respectively, and therefore very convincing in our eyes.

To further highlight the validity of our observations we have also introduced a new Figure S9, showing additional statistical analysis of raw N-K EELS data using Blind Source Separation (BSS) algorithms as implemented in the Hyperspy code (<https://10.5281/zenodo.7263263>). BSS, a machine learning technique widely used across many fields dealing with spectroscopy data, allows for the unmixing of original source signals from their intermixed observations (<https://doi.org/10.1016/C2020-0-01790-1>). The additional analysis presented here shows subtle but clear nanoscale variations of the N K fine structure within the organic grain, especially at energies above 400eV. The peak at 405.9 eV is particularly prominent in one of the two statistically identified components, which further highlights that the observed features are real and not processing artefacts, and that our approach does in fact provide the ability to observe spatial variations on the nanoscale of the identified N-containing spectral bands.

Furthermore, in all previous work, N-K edge extended fine structure above 400 eV rarely showed any bands by EELS. The bands detected here, while noisy, are certainly above the background level in contrast to previous work e.g., Fig. 3e of the aforementioned Alexander et al. 2017 paper.

Data from compound-specific mixtures to establish detection limits and preservation through exposure to the x-ray, ion and electron beam exposures are needed to provide adequate confidence in the detection of amino acids and nucleobases.

It is certainly true that it would be desirable to acquire EELS spectra over a whole range of N-bearing reference standards that would also cover the complex chemistry of extraterrestrial OM. But this is far beyond the scope of the present paper and would require a complete research project of its own. See, for example, the reference papers of Leinweber et al. (2007) or more recent work about core-loss C-K edge EELS on polymers (Colby et al. Ultramicroscopy 2023) which showcase such extensive work.

Perhaps more crucially, we would like to stress that meteoritic OM is a highly complex material, as also clearly evidenced by data in Figure S9 discussed in the response to the previous question. Consequently, the positions of relevant bands may not fit perfectly to the assignments from reference papers (e.g., Leinweber et al. 2007), which will have been acquired on pure 'standards', in highly controlled conditions. Therefore, band positions of N-containing compounds identified here are likely spread over a widened energy range due to local heterogeneity, rather than restricted to exact values from the literature.

Acquiring a range of pure standards, or "mixture standards", while worthwhile for the community to, e.g., explore the detectability and beam sensitivity of given compounds using monochromated STEM-EELS, would only serve here to reproduce broad band assignments that are readily found in the literature, without providing further interpretability of the present data.

In the updated version of the manuscript, we have modified Fig. 8 and used small windows over respective band positions to make the assignments and their slightly wide-band nature clearer to the reader.

The soluble organic compounds are at much lower abundance, and are much more susceptible to damage than the insoluble compounds, and thus the reported detection is not convincing.

We would argue that it is in fact not really known what concentrations of SOM and IOM are present in organic matter that is observed in polished sections of extraterrestrial samples. This is because those two assignments are based on work carried out on extracted materials by acids and solvents, the very issue that the present work is aiming to avoid by studying less invasively processed samples.

In organic grains within meteorites, for example, nitrogen concentrations may vary from below detection limit to over 10 at. %, and it is likely that N-containing compounds with soluble AND insoluble character contribute to this signal. The fact that the soluble components are more susceptible to damage is an even better argument for analyses like the one presented here on minimally processed materials, because it is likely that less material is lost compared to other work using solvent extraction.

We have added a sentence to the introduction to make this argument and further motivate the use of minimally processed samples for this kind of analysis.

Better presentation of the data processing, e.g., exactly what PCA and smoothing parameters were used, and use of analog standards for validation could substantially improve the manuscript.

Please see our comments above about the PCA smoothing parameters: a new section on data processing was added to the Supplementary Information, alongside additional statistical analysis of the data using machine learning and blind source separation techniques. While we agree that the availability of standards would be desirable, not only would this require a research project of its own, as stated above, the in-depth processing provides in our eyes a far more convincing demonstration of the reliability of the presented data. This is because meteoritic OM is a complex material, and even the analysis of reference standards and pure substances (of amino acids, for example) would not be directly comparable to the spectra reported here.

Reviewer #2

The water content of the meteorite would be one reference for estimating the history of aqueous alteration (ref. 43). In this sense, comparative feedbacks with Bates et al. 2023 & Daly et al. could be discussed complementary.

We do cite the Bates et al. (2023) paper and also refer to that paper in the relevant section, although it is not directly relevant to our paper because it reports on bulk properties of Winchcombe. Because we do not aim at nor provide data relevant to a comprehensive discussion of aqueous alteration effects on Winchcombe, we only refer to that work in one paragraph and feel an extension of this discussion would be too speculative and would unnecessarily lengthen the "Discussion" section of the manuscript.

We thank the reviewer for drawing our attention to the work of Daly et al. (2023). Despite our best efforts, we could not find any published or pre-print version of this paper, and we therefore believe it is not published yet and is still under review at the time writing. We would be happy to add a reference to it if it becomes available in time for the publication of our work.

Based on the non-destructive measurement, the authors discuss indigenous SOM (refs. 44, 45) and IOM profiles (e.g., STXM data interpretation) throughout the paper. In particular, I would add the need for verification of consistency with the latest findings on IOM to feedback on the present data.

We have added the suggested reference to Sephton et al. (2023) about IOM investigations on Winchcombe and also more explanations. However, that work did not specifically discuss N-containing compounds, but rather related the general makeup of Winchcombe IOM to other types of chondrites confirming that it is a CM chondrite. Because it is impossible to say how exactly IOM and SOM relate to the organic matter described in this work (which is why we refer to this material simply as "OM" or "organic grains"), direct comparison to other work which deals explicitly with either SOM or IOM extracts, is difficult.

I think that that the early view literature describing the process of survey, sampling, initial evaluation, and curation immediately after the Winchcombe meteorite fall is an important paper in terms of assuring the value of the sample.

We thank the reviewer for this suggestion. We have added the paper by Russell et al. (2023).

The authors mention nitrogen (N)-containing heterocyclic compounds in samples returned from carbonaceous meteorites and carbonaceous asteroids (refs. 24-27, 48). Their point of view is very interesting for astrochemistry and prebiotic molecular evolution. On the other hand, however, it may be necessary to provide additional explanation as to whether or not the signal attribution of imino groups (e.g., -C=N-) derived from heterocyclic compounds can be evaluated with any degree of accuracy in the present spectroscopic measurements. In fact, nitrogen (N) exists in various chemical forms such as ammonia (NH₄⁺), amine (-NH₂), imine (-C=N-), or nitrate (-NO_x) forms. Would it be possible to show an organized (e.g., table data) list of the raw values of the spectral chart? I would like to suggest that the key points of the qualitative attribution of this nitrogen signal be described in the results of the main text or in the SI script.

It is true that C-N double bonding environments (i.e., imine functional groups) could also derive from the presence of N-heterocycles such as imidazole. We appreciate the suggestion of the reviewer and note that, in this instance, they would show absorption features at higher energies than we discuss. We have added a comment to clarify this point in the relevant paragraph (line 302-304).

More generally, it is true that nitrogen occurs in various forms, as the reviewer points out. We have therefore added a new Table S1 containing the main characteristic bands of the extended fine structure in the N-K edge we observe in our data in the SI file, and we discuss these and contrast to different forms of nitrogen in the text, for example, the presence of nitro groups.

Reviewer #3

Personally, I would like to see hydrogen, carbon and nitrogen isotopic compositions of C-rich area in FIB sections, especially ROI containing imine and nitrile functions. Those isotopes may help to decipher nature, origin and evolutionary history of organics having imine and nitrile functions. If authors have a chance to access to NanoSIMS, please conduct isotope imaging analyses. This improved the manuscript.

We agree with the reviewer that such information would be extremely valuable, but as they point out, this would only be possibly using different techniques such as NanoSIMS. SIMS analyses of organic grains are very important, specifically concerning anomalous organic hotspots or coldspots with respect to their specific functional chemistry. However, work on Winchcombe is an ongoing research project, and while we indeed plan to conduct NanoSIMS analyses, they are both time-consuming and mostly independent of the STEM-EELS and STXM work presented here, and will fall within the scope of a separate publication.

It is also true that we could analyze our FIB sections by NanoSIMS, which is geometrically challenging, but possible. However, this would destroy the already very thin lamellae, and it would preclude any further EELS data acquisition on them. The work presented here aimed specifically at the functional chemistry of the Winchcombe OM by high-energy-resolution EELS.

I would appreciate if authors could describe more details about FIB methodology to the Winchcombe even author cite a proper reference.

We have added details on the FIB preparation in the Methodology section of the paper.

One additional point would be the asteroid Bennu (CM-type by spectroscopic study on board) samples by NASA O-Rex mission. I think that it would be nice to mention possible genetic link to asteroid Bennu even a little bit speculation.

We have added a sentence concerning the Bennu samples and prospects for similar analyses in the last paragraph.

If I remember correctly, a STXM is not sensitive to nitrogen. It requires a few wt% to detect N-bonding peaks with proper signal-to-noise ratio.

This is true, which is why STXM N-K edge spectra are typically noisy. This is also why we used EELS to extract more information and we focused our analysis of the N-specific data on the EELS results (while comparing the C K data obtained across both techniques).

Please see reference Zubavichus et al (2005, Innershell Absorption Spectroscopy of Amino Acids at All Relevant Absorption Edges. J. Phys. Chem. A, 109, 6998-7000) for amino acid study with STXM C and N-XANES. There is a combination of C and N-XANES spectrum in each amino acid.

Thank you very much for this suggested reference. We have included it in the reference list and also refer to it in our discussion concerning amino acid detection and N-heterocycles such as imidazole. It is still very difficult to directly compare such reference spectra of pure substances with extraterrestrial organics, specifically because the C-K edge fine structure is dominated by the polyaromatic kerogen.

P5, L106: It is better to have a proper reference for "because aliphatic functional groups are easily disturbed by alteration reactions". Alteration reactions of what?

We have specified the alteration reactions and also added references to Ito et al. (2022) concerning labile aliphatics bonding environments in Ryugu OM and to Vinogradoff et al. (2017).

Please clarify this statement of "the STXM analyses did not modify the OM of those lamellae". C-XANES of FIBW-02 by STXM and that of EELS look different to me, for example, (1) the

ratio between 285.0eV and 286.5eV, and (2) no 286.6eV peak (or just a shoulder?) in FIBW-02 by STXM but that in FIBW-02 by EELS (indicating as arrow in Fig. 6). Could this be some modification during electron and/or photon irradiation? Please see reference “Wang et al., 2009, Radiation damage in soft X-ray microscopy. *Journal of Electron Spectroscopy and Related Phenomena* 170 (2009) 25–36”.

We certainly agree with the reviewer that observation with STXM can, and does lead to radiation damage to samples, as we state on page 11 of the manuscript.

However, in this specific case, the observed differences may not be due to prior beam damage during beamline observation. We believe that while it is true that the C-K edge spectra of FIBW-02 by STXM and EELS look different, this is because they have been acquired on slightly different areas of that very same lamella. Spatial resolution of the STXM analyses is worse than by EELS, and the observed differences underline again the chemical complexity of this material on a sub- μm scale and the need to carry out complementary observations with high-resolution monochromated STEM-EELS, as well as the value of this technique for the field. This can be further highlighted by the C K EELS edges acquired from different parts of the FIBW-02 lamella plotted in Figure S2 of the Supplementary information file. This variability is discussed on page 6 of the manuscript, within the “Organic matter C K-edge functional chemistry” section, where the more localized analysis enabled by STEM-EELS is highlighted to FIBW02.

Furthermore, spectra from lamellae FIBW-07 – FIBW-09, where only EELS has been acquired, look very similar to those spectra where both techniques have been applied in succession. We have added some more explanations concerning that issue.

In fact, from my experience of C-XANES analysis of Ryugu FIB samples, there were obvious modification in 286.6eV peak (aliphatic) by photon irradiation during STXM analysis. I think that it is better to describe photon and electron doses during your analysis with STXM and EELS.

As mentioned above and pointed out by the referee, the aliphatics peak is indeed prone to alteration by irradiation. However, our discussion mainly centers around the aromatic and the C-O ketone bonding environments, which seem to be more robust as no modification to its fine structure was observed in our experiments in spite of repeated data acquisition in some areas of the lamella.

The exact electron dose can vary from dataset to dataset, as it is dependent on the size of the scanned area and the duration of the spectrum image acquisition. Nevertheless, the order of magnitude of the electron dose remains similar across all our work. The probe current was restricted to approximately 3-5 pA using the monochromating slit, providing a compromise between energy resolution (70-90 meV, as estimated using the full-width at half-maximum of the zero-loss peak) and signal level. This current was delivered to the sample in a probe of 0.1 nm diameter in size, rastered across large regions of the sample in a sub-sampled way (that is to say, the distance between pixels in a dataset was larger than the probe size: we used typically 1 nm pixels or larger) – which is one of the experimental procedures designed to mitigate possible beam damage (so if damage does occur, the neighbouring pixel is far enough away that delocalized damage to the sample is limited). For a typical dataset, the electron dose per spectrum image is $3\text{-}5 \cdot 10^4 \text{ e}^-/\text{\AA}^2$.

REVIEWERS' COMMENTS

Reviewer #1 (Remarks to the Author):

Given the quality of the data and the potential for further advancement, I now recommend publication with some minor revisions, as described below.

Ln 24 :“ extracted by the a similar harsh extraction process” This is a mischaracterization of the SOM and IOM extraction processes. The SOM and IOM extraction processes are distinct, the former involving solvents and temperatures known not to alter the amino acids of interest, and the later has been established to be minimally altering to the IOM components, with the acids primarily attacking the mineral matrix. There is no need to use qualitative, pejorative terms such as “harsh” to demonstrate that FIB extraction can preserve soluble components and petrographic context lost in other methods. The solvent extraction methods are complementary and the only way to establish bulk composition. I suggest the following:

Insoluble organic matter (IOM) can be extracted from the bulk sample by acid-dissolution, which results in a demineralized, concentrated organic residue for further studies^{10,11}. A third approach, that permits local measurement of soluble and insoluble OM within the original petrographic context is...

Ln 29-30 This is again a mischaracterization of the extraction process as altering of the OM, which the authors have not shown and is in contrast to what is documented in the literature, followed by an overgeneralization of the FIB extraction processes as doing no damage. The amount of damage to the organics during the FIB extraction process is highly dependent on the exact procedure followed, and on details of the sample that are not known a priori, e.g., SEM beam exposure, conductivity of organic/matrix, and specific susceptibility of the organics in the sample.

In regards to the FIB damage, the methods section now provides extensive detail of for the FIB extraction process, but is missing a couple of crucial pieces of information. It is stated that the lamella are examined by SEM at voltages below 5 kV prior to STXM. Since the threshold for knock on damage for H in organics is around 1 kV, 5 kV may not be low enough. This could explain the low abundance of aliphatic and carboxyl functional groups reported here. However, it is not just the voltage but the dose that matters. The dose required to obtain a quick, low magnification documentation image at 5 kV to enable the STXM and EELS work might have no detectable effects (depending on the sample). However, It is also possible that the 1 kV Ar⁺ ion beam processing could cause thermal damage and loss of the aliphatic and COOH groups, if this milling was done at room temperature. The authors should clarify that detail. At present, there is not a definitive way to determine exactly how much damage the FIB processing might have caused to the samples in this study, although it is clear from the retention of peaks besides the aromatic, it is not severe. The authors have made good efforts to minimize the possible damage, but not all parameters are ideal. The best solution is to add the missing details, not overstate the pristinity claims, and leave it to future studies to confirm or negative the negligible alteration.

For the lines 129-30, I recommend that the authors just stick to saying that “With this approach, it is possible to investigate the petrographic context of the OM as unaltered as possible and potentially detect both soluble and insoluble components”, which is indisputable and avoids making unsubstantiated claims about the lack of alteration to the organics by the FIB or dramatic alteration by the solvent extraction.

Ln 33-34 "The assignments to either category..." This statement is out of place and attempts to pre-empt the question of whether the assignments of particular spectral features to soluble components is probable, given the bulk relative abundances (bulk studies show that the soluble component is much less abundant than the insoluble). I recommend leaving the question of assignment probabilities to the discussion of the results.

Overall, the N EELS data do present a significant advance over prior measurements, particularly with regard to the N heterocycle species. However, the assignment of features to amino acids is speculative. If, as the authors claim, the exact energies of the N functional groups can move around, and they do not need to match reference spectra exactly, this widens the potential range of other assignments to the same features. The other problem is that the relative intensity of the ~ 401 eV features to the broad feature at ~ 405 to 410 eV, and the feature width in the 405 to 410 eV range, is in disagreement with the cited reference spectra (ref 42, Fig. 3). The authors might be correct in their assignment, but the data are far from a definitive detection. My recommendation on this is simply to soften the language regarding this claim. It is a decent tentative assignment. Use the terms, tentative and "suggestive of" with the assignment, and it will still remain interesting and publication worthy.

Lastly, to address the prominent feature in the STXM at ~ 288.5 that is missing or dramatically less pronounced in the EELS data, I would comment that there is recently recognized issue with several synchrotron STXM instruments that causes an artifact at this energy range. During the Hayabusa2 initial analysis of IOM, researchers measuring the same sections at Spring8 and ALS noticed this artifact. Second order features from oxygen absorption can appear at this energy at synchrotrons with a design that does not explicitly include a filter to exclude them. This shows up as a "shelf" in the 288eV to 289 eV range. It is particularly prominent in FIB sections in which the organics are mixed with O-rich minerals. It can be reduced to some degree through the use of N₂ filter (gas backfill in the chamber) during C STXM collection. I am not specifically aware of any publication that discusses this further, but it was something David Kilcoyne took into account for the design on ALS 5.3.2 and was also integrated into the CLS, but is not incorporated into Diamond or Spring8. It does not result from damage to the sample by the x-ray beam--- which would be seen in subsequent EELS measurements. Hikaru Yabuta the Hayabusa2 IOM team lead can likely provide more information if needed, or possibly Harold Ade who helped design ALS 5.3.2. Unfortunately David Kilcoyne passed away before this made it to print.

Reviewer #3 (Remarks to the Author):

This is my second review of this manuscript dedicated to the functional chemistries in nitrogen-bearing organics of Winchcombe meteorite by Vollmer and colleagues.

The authors have addressed comments by myself and I'm happy to recommend publication in Nature communications. I only have minor comments below (I used 441325_1_merged_1699275949.pdf).

1. title: I think that it is better to delete high-spatial resolution from the title. Just an idea to emphasize importance of nitrogen functional chemistry of Winchcombe organics.

2. Introduction: It is better to have a proper reference after "...of which a significant fraction is organic matter (OM)"

3. I am bit confused that you wrote "as well as ultra-high energy resolution ($\Delta E < 0.1\text{eV}$ for both STXM and EELS)" in P5. However, you wrote an energy resolution of STXM (0.1 eV or 0.2-0.5eV in P21, Scanning Transmission X-ray Microscopy). If I remember correctly, an EELS system does not allow to acquire such a high energy resolution like 0.1eV, in general it is 0.3eV which is equivalent to FE-gun nominal energy spread. But I may be wrong.

Date 21.12.2023

Reviewer #1

Ln 24 :” extracted by the a similar harsh extraction process” This is a mischaracterization of the SOM and IOM extraction processes. (...) I suggest the following: Insoluble organic matter (IOM) can be extracted from the bulk sample by acid-dissolution, which results in a demineralized, concentrated organic residue for further studies^{10,11}. A third approach, that permits local measurement of soluble and insoluble OM within the original petrographic context is...”

We agree with the reviewer`s suggestion and apologize for this qualitative statement. We have adopted the suggested changes in the manuscript.

Ln 29-30 This is again a mischaracterization of the extraction process as altering of the OM, which the authors have not shown and is in contrast to what is documented in the literature, followed by an overgeneralization of the FIB extraction processes as doing no damage. (...)

We have adopted the suggested change.

In regards to the FIB damage, the methods section now provides extensive detail of for the FIB extraction process, but is missing a couple of crucial pieces of information. (...) The best solution is to add the missing details, not overstate the pristinity claims, and leave it to future studies to confirm or negative the negligible alteration.

We agree, and we should have been more concise: the FIB process (or any sample preparation) can never be completely devoid of damage. Not only are dose and voltage crucial for keeping the samples as unaffected as possible, it is indeed likely that H atoms are the first to be lost – although not completely as the ELNES fine structure observed would otherwise not contain those spectral bands identified as corresponding to specific functional groups.

We have added details on the Ar treatment to the FIB Methods section, as it has been widely shown to result in quantifiable improvement in sample surface quality and in a reduction of Ga-exposure-related damage, certainly in the field of semiconductors. Relevant references have been added – and we note that further exploration of these effects for organics, and the use of cry-FIB, as the referee suggests, may lead to further improved methodologies.

We also detailed in the Methods section how SEM exposure was specifically minimized to avoid electron beam exposure effects following the final thinning steps.

Ln 33-34 *"The assignments to either category..." This statement is out of place and attempts to pre-empt the question of whether the assignments of particular spectral features to soluble components is probable, given the bulk relative abundances (bulk studies show that the soluble component is much less abundant than the insoluble). I recommend leaving the question of assignment probabilities to the discussion of the results.*

We have deleted this statement.

Overall, the N EELS data do present a significant advance over prior measurements, particularly with regard to the N heterocycle species. However, the assignment of features to amino acids is speculative. If, as the authors claim, the exact energies of the N functional groups can move around, and they do not need to match reference spectra exactly, this widens the potential range of other assignments to the same features. The other problem is that the relative intensity of the ~ 401 eV features to the broad feature at ~405 to 410 eV, and the feature width in the 405 to 410 eV range, is in disagreement with the cited reference spectra (ref 42, Fig. 3). The authors might be correct in their assignment, but the data are far from a definitive detection. My recommendation on this is simply to soften the language regarding this claim. It is a decent tentative assignment. Use the terms, tentative and "suggestive of" with the assignment, and it will still remain interesting and publication worthy.

We are absolutely aware of the fact that detected amino acid bands are rather tentative. In the modified manuscript we have softened our language accordingly. We stress again, however, that the shapes of spectra from pure substances as in the Leinweber et al. paper (Ref 42) cannot be directly compared to those of complex OM in meteorites.

Lastly, to address the prominent feature in the STXM at ~ 288.5 that is missing or dramatically less pronounced in the EELS data, I would comment that there is recently recognized issue with several synchrotron STXM instruments that causes an artifact at this energy range. (...)

This comment is difficult to address, because we argue in the paper on the contrary that STXM and EELS data are rather similar in that respect and that the 288.5 feature is not that very prominent in both STXM and EELS. Even more so, the fact that we do not see this feature in EELS, where no such instrumental artefact is expected, means that this is new information that STXM would not have been able to uncover. Our results therefore suggest the low signal intensity at 288.5 eV is 'real'. In any case, this comment is also difficult to adopt because there is no reference we could cite, and it is a very recent observation and further work is needed to confirm or refute this observation. We would therefore simply leave our argumentation as it is.

Reviewer #3

1. title: I think that it is better to delete high-spatial resolution from the title. Just an idea to emphasize importance of nitrogen functional chemistry of Winchcombe organics.

We would still like to stick to our original title to emphasize the fact that we performed our analyses in minimally processed samples in contrast to functional chemistry observations on bulk and extracted samples.

2. Introduction: It is better to have a proper reference after "...of which a significant fraction is organic matter (OM)"

We have shifted the references from the following sentence to this position.

3. I am bit confused that you wrote “as well as ultra-high energy resolution ($\Delta E < 0.1\text{eV}$ for both STXM and EELS)”in P5. However, you wrote an energy resolution of STXM (0.1 eV or 0.2-0.5eV in P21, Scanning Transmission X-ray Microscopy). If I remember correctly, an EELS system does not allow to acquire such a high energy resolution like 0.1eV, in general it is 0.3eV which is equivalent to FE-gun nominal energy spread. But I may be wrong.

The referee is correct in that the native energy spread of a cold field emitter (CFEG) is indeed $\sim 0.3\text{eV}$. However, as explained in the methods section and in the main text, our results were acquired in a specialized STEM instrument, which is equipped with an advanced monochromator for high resolution EELS measurements (with achievable resolutions currently $\sim 5\text{meV}$ at 60kV acceleration voltage: see for instance N. Dellby et al., *Microscopy & Microanalysis* 26, S2, 1804-1805, 2020).

However, rather than the record 5meV our instrument is capable of, the experiments here were performed at 50-90meV energy resolution as set by the monochromator selection slit, which matches the widest expected spectral feature in this range (due to a range of physical effects, such as lifetime broadening). This choice also offers an optimized balance between energy resolution and beam current, to help with signal detectability.